# Effects of Growing Rod Technique with Different Surgical Modes and Growth Phases on the Treatment Outcome of Early Onset Scoliosis: A 3-D Finite Element Analysis

**DOI:** 10.3390/ijerph19042057

**Published:** 2022-02-12

**Authors:** Baoqing Pei, Da Lu, Xueqing Wu, Yangyang Xu, Chenghao Ma, Shuqin Wu

**Affiliations:** 1Beijing Key Laboratory for Design and Evaluation Technology of Advanced Implantable & Interventional Medical Devices, Beijing Advanced Innovation Center for Biomedical Engineering, School of Biological Science and Medical Engineering, Beihang University, Beijing 100083, China; pbq@buaa.edu.cn (B.P.); luda_bme@buaa.edu.cn (D.L.); youngerxv@buaa.edu.cn (Y.X.); ytmch888@163.com (C.M.); 2School of Big Data and Information, Shanxi College of Technology, Shuozhou 036000, China

**Keywords:** children’s health, early onset scoliosis, bilateral/unilateral posterior fixation, different growth phases, finite element analysis

## Abstract

Early onset scoliosis (EOS) is emerging as a serious threat to children’s health and is the third largest threat to their health after myopia and obesity. At present, the growing rod technique (GRT), which allows patients to regain a well-balanced sagittal profile, is commonly considered as an invasive surgical procedure for the treatment of EOS. However, the risk of postoperative complications and instrumentation breakage remains high, which is mainly related to the choice of fixed mode. Several authors have studied primary stability and instrumentation loads, neglecting the mechanical transmission of the spinal long-segment model in different growth phases, which is fundamental to building a complete biomechanical environment. The present study aimed to investigate the kinematic and biomechanical properties that occur after GRT, across the long spinal structure and the posterior instrumentation, which are affected by unilateral or bilateral fixation. Accordingly, spinal segments (C6-S1) were loaded under flexion (Flex), extension (Ext), left lateral bending (LB), right lateral bending (RB), left torsion (LT), and right torsion (RT) using 11 established spinal models, which were from three growth phases. The stress distribution, spinal and intervertebral range of motion (ROM), counter torque of the vertebra, and bracing force on the rods were measured. The results showed that bilateral posterior fixation (BPF) is more stable than unilateral posterior fixation (UPF), at the expense of more compensations for the superior adjacent segment (SAS), especially when the superior fixed segment is closer to the head. Additionally, the bracing force of the instrumentation on the spine increases as the Cobb angle decreases. Accordingly, this biomechanical analysis provides theoretical suggestions for the selection of BPF or UPF and fixed segments in different growing phases.

## 1. Introduction

EOS is a three-dimensional spinal deformity that occurs in children before the age of 10 years old and is mainly caused by congenital spinal deformities, neuromuscular diseases, scoliosis-related syndromes, or idiopathic causes [1,2]. During the rapid growth and development that children experience, spinal deformities can affect the physical health and cardio-pulmonary function of a child, leading to death from cardiac failure in severe cases [3]. Conservative therapy is more suitable for mild scoliosis and is greatly reliant on patient compliance, and results in inchoate spinal correction [4]. Growing rod techniques, such as BPF and UPF, are widely used as a no-fusion procedure for severe scoliosis [5,6]. However, the risk of postoperative complications and instrumentation breakage is still high, mainly related to the choice of the fixed mode of stabilization.

The recent literature contributed to a better understanding of the basic biomechanics behind GRT. There is a smaller wound and less restriction after UPF than BPF, while UPF has much weaker stability. On the contrary, BPF can cause higher rates of postoperative infection [7,8]. Computational studies based on finite element analysis have also reported similar findings [9,10]. On the other hand, more frequent instrumentation extensions may lead to a higher risk of rod failure but may provide greater growth potential for the spine, since this method promotes development with periodic scoliosis support [11,12].

Older publications rarely addressed the biomechanical reasons for the different surgical effects between UPF and BPF. Clinical investigations are common for evaluating therapeutic effects and rely on the statistical method. Long-term follow-up patient data make it difficult to adapt to medical improvements and medical needs. The finite element method (FEM) may be more suitable for the current situation, avoiding the shortcomings of an insufficiency of experimental specimens, low repeatability, and low biological simulation. Well-known intervertebral disc (IVD) pressure measurements have often been used as an index of the loads transferred through the anterior column of intact and instrumented spine segments, following the destabilization of one or more functional spine units (FSU), due to the damage of the bony structure [13,14,15]; however, it neglects the biomechanical contributions of the remaining bony structures and ligaments. Other studies relying on biomechanical experiments have the difficulty of obtaining a human specimen from children, as only spines with one growth phase and a specific profile can be obtained, thus, neglecting the significance of spinal growth and the internal environment of the spine. Additionally, many studies abandoned the long-segment spine due to the complexities related to reconstruction and computation.

No prior study has ever investigated the biomechanical mechanism between the 20 spine segments and the instrumentation following UPF or BPF, nor have the investigated models from one patient with multiple growth phases. Therefore, the current study aimed to interpret and understand the reasons for the occurrence of spinal complications. In particular, the present study focused on variations in biomechanics due to GRT fixation, and the stress distribution, spinal and intervertebral ROM, counter torque of the vertebra, and bracing force on the rods, all of which were measured in this study. A more practical motivation for this study is to provide a theoretical guide for the rational selection of a surgical strategy.

## 2. Materials and Methods

The research was approved by the Science and Ethics Committee of the School of Biological Science and Medical Engineering at Beihang University (protocol code: BM201900125).

### 2.1. Subjects

The geometry of the spine and growing rods were constructed from a patient with EOS treated using the BPF surgical procedure. The patient (9 years, 115 cm, 30 kg) did not have any other known musculoskeletal disorders and underwent three surgical operations. The preoperative X-ray displays a thoracic curve Cobb angle of 62.9°/54.4°/29.9° (thoracic apex: T8; thoracic cephalic vertebra: T5; thoracic caudal vertebra: T10) and a lumbar curve Cobb angle of 45.1°/44.1°/29.7° (lumbar apex: T12; lumbar cephalic vertebra: T11; lumbar caudal vertebra: L3) with sagittal profiles, which were captured at an interval of 1 mm and at a resolution of 512 × 512 px using a CT scanner (SIEMENS/SOMATOM Definition) from Germany. Three CT images and three reconstructed models are shown in Figure 1.

### 2.2. Creating the Base Model

To better understand the context of the article, a flowchart of the entire study is shown in Figure 2. The initial 3-D bony model reconstruction was developed using MIMICS (version: 17.0; company: Materialise; location: Europe Belgium). The HU values were set to 226~1262 and the smoothing factor was adjusted to 0.6. The marginal conformations caused by HU values were eliminated with a reconstructed vertebra unit. Point clouds were further smoothed using Geomagic Studio (version: 2013; company: Geomagic; location: Triangle, NC, USA) to facilitate conversion into a CAD model with surface slices. The intervertebral disc (IVD) and growing rod instrumentation models were built by sweeping and lofting operations in SolidWorks (version: 2019; company: Dassault Systemes SE; location: Paris, France). Structures such as ligaments were established in simulation software. Finally, post-processing process was carried out in MATLAB (version: R2018b; company: Mathworks; location: Natick, MA, USA).

To evaluate how the BPF and UPF affect the load and movement of the spine, a total of 11 spinal models based on the CT from three growth phases have been reconstructed in Figure 3, within one pre-operative model for the control group. The results reveal the effect of different phases and different surgeries on the spine from two aspects.

### 2.3. Adopted Mesh and Material Properties

Bony structures and soft tissues were discretized to the tetrahedral (C3D4) element type and hexahedral (C3D8) element type, respectively, avoiding the time-consuming phenomenon in the hexahedral discretization of complex bony structures. This grid division method ensured the high strain characteristics of soft tissues and thus allowed the finite element model to achieve high accuracy and reliability of the finite element model. Ligament tissues were divided into the anterior longitudinal ligament (ALL), posterior longitudinal ligament (PLL), interosseous transverse ligament (ITL), capsular ligament (CL), interspinous ligament (ISL), supraspinous ligament (SSL), and ligament flavum (LF). These ligaments were modeled with three-dimensional truss elements and were discretized in a one-dimensional line grid. The Cobb angle of the growing rod in FEM is the same as the Cobb angle in the patient’s body. All vertebral bodies were perforated using SolidWorks in Boolean operation. Then, the surface of the hole and screw were subsequently tied. Each structural material property is presented in Table 1.

### 2.4. Boundary and Loading Conditions

Soft tissues such as muscles are hard to reconstruct through the use of FEM due to the alterability and nonlinear characteristics of muscles. Thus, the muscle force and pressure on the vertebra are the following load, which is a transitive force along the curve of the spine. Additionally, the inferior extremity of S1 was restrained in all degrees of freedom and a torque and displacement load were applied to the superior extremity of C6 (Figure 4a). In the follow-up study, the growing rod was braced at 12 mm to obtain the force transferred from the rod to the vertebra.

Every vertebra was created as a wedged cylinder consisting of cortical and cancellous bone and vertebra with a posterior structure. The internal IVD component divisions were as follows: fibrous annulus (AF), nucleus pulposus (NP), and cartilage endplate. The vertebrae were connected with ligaments and were bonded to the IVD surface (Figure 4b). The friction coefficient of friction between the facets was set to 0.01. A torque of 7.5 Nm and an angular displacement of 1 rad were set.

The realization of the following load set is a relatively vital process. Usually, the action mode of the following load can be replaced with addition to a thermosensitive truss. By assigning a temperature field change to a thermosensitive truss, the mechanical load can be transferred among the elements in the form of thermal load using the thermal expansion effect. Thus, the direction of the load will be axially transferred along each vertebral center. The following load of each vertebra is set on the basis of Pasha’s study [24] and is shown in Table 2.

The expansion coefficient α was defined for the prestressed element, and the unbonded prestress was determined using the following formula [21]:*δT* = ε/α(1)
where *δT* is the thermal load for the iteration, ε is the thermal strain (growing strains) for the iteration, and α is an arbitrary number representing the thermal expansion coefficient.

### 2.5. Validation of the Model

Model validations are directly related to the results of subsequent studies and conclusions and are generally performed with one FSU. Direct validation compares the finite element results with the biomechanical data from the cadaver. Additionally, indirect validation compared the finite element results with other literature data. Some researchers have conducted biomechanical experiments on pediatric bones [25,26,27] providing us with a reference basis.

The FSU of C6-C7 was taken to validate the model fidelity according to the experimental conditions proposed by Luck [27]. In the flexion and extension simulations, all of the degrees of freedom were restrained for C7, and C6 was imposed with the moment values of ±0.5 Nm, ±1.0 Nm, ±1.5 Nm and ±2.0 Nm. The angular displacement of C6 in children is shown in Figure 5. The simulation results were consistent with the experimental results under both flexion and extension conditions. The average error between the simulations and the experiment was for the flexion and extension conditions, which showed errors of 5.9% and 6.7%, respectively. The Cobb angles of the model are quite close to the CT Cobb angles throughout the three growth phases.

## 3. Results

### 3.1. IVD Stress of SAS and Overall ROM of Spine under Torque Loading

T1-L4 fixed: In the intact condition, the SAS IVD stress showed a unified trend in six loading modes (BPF > UPF > PREO) (Figure 6a). No significant differences were noticed among the different loading modes. On the contrary, the overall spinal ROM followed the following trend for each of the different models: BPF < UPF < PREO (Figure 6b). The instrumentation also demonstrated obvious LB ROM restrictions (declined by 72.97% after UPF, declined by 91.89% after BPF), but much lower ROM restriction on Flex (declined by 27.58% after UPF, declined by 65.51% after BPF).

T2-L4 fixed: In the intact condition, the LT IVD stress was higher than other loading modes, and the following trend was maintained: BPF > UPF > PREO (Figure 6c). On the contrary, the overall spinal ROM followed the following trend for each of the different models: BPF < UPF < PREO (Figure 6d). Following GRT, the ROM was consistent for six loading modes with a varying degree of decline. Among them, the ROM of LB declined by 72.46% (UPF) and 84.05% (BPF), but much lower degrees were observed for Flex (declined by 29.63% after UPF, declined by 62.96% after BPF).

T3-L4 fixed: In the intact condition, the LT IVD stress was higher than that of the other loading modes, while retaining the following trend: BPF > UPF > PREO (Figure 6e). The ROM trend was the same as that of the above two groups, as follows: BPF < UPF < PREO (Figure 6f). Following UPF, the ROM of LB and Flex declined by 69.56% and 25.93%, respectively. Additionally, BPF decreased to levels of 81.16% and 50.00%.

The model of the T2-L4 segments, fixed by the instrumentation, is shown separately (Figure 7).

Adjacent disc stress: The maximum stress was mostly concentrated at the left anterior part of the IVD and was obvious in the LT loading mode (about 5 MPa in PREO and UPF, and 6 MPa in BPF). However, it is evident that the stress concentration in the red regions is smaller than that in the other regions. On the contrary, the stress shown in the light green area more closely approximates the average IVD stress (about 2.08 MPa~2.92 MPa in PREO and UPF, and 2.5 MPa~3.5 MPa in BPF).

ROM: The maximum ROM in the three groups was 2.1 rad, 1.0 rad, and 0.70 rad, respectively. After UPF or BPF, the spinal activity capacity was much smaller than PREO, mostly affecting the LB and RB, which showed a larger difference compared to other loading modes.

### 3.2. IVD Stress and ROM of SAS under Angular Displacement

T1-L4 fixed: In the intact condition, the SAS IVD stress showed a unified trend in the six loading modes (BPF > UPF > PREO) except for in the RT group. Among them, the IVD stress in Ext was obvious (about 21.25 MPa of BPF, 16.32 MPa of UPF), but the most distinct difference was in LB (increased by 4.18 times after UPF and 2.95 times after BPF) (Figure 8a). Following GRT, the ROM of SAS was significantly increased due to the SAS compensating for the activity of the fixed segments, leaving the following trend: BPF > UPF > PREO. This is different from the trend that was observed during torque loading. The maximum compensation ratio occurred in LB (increased by 3.53 times after BPF and 2.00 times after UPF) (Figure 8b).

T2-L4 fixed: In the intact condition, the SAS IVD stress showed a unified trend in the six loading modes, as follows: BPF > UPF > PREO. Among them, the IVD stress of RB and LT was larger than that of the others (about 14.5 MPa of RB after BPF, and 5.20 MPa of LT after UPF), but the IVD stress ratio increased and was significant in RB (about 8.57 times after BPF and 2.14 times after UPF) (Figure 8c). In the ROM diagram, the values all increased, leaving the following trend: BPF > UPF > PREO. LB showed the greatest increase in the ROM (about 32.5 times after BPF and 6.5 times after UPF) (Figure 8d).

T3-L4 fixed: In the intact condition, the SAS IVD stress showed a unified trend in the six loading modes, as follows: BPF > UPF > PREO. Among them, the IVD stress of LB and RT was larger than the others (about 16.10 MPa of LB after BPF, and 12.05 MPa of RT after UPF), but the IVD stress ratio increased and was significant in LB (about 4.08 times after BPF and 1.77 times after UPF) (Figure 8e). In the ROM diagram, the values all increased and followed the trend (BPF > UPF > PREO). LB showed the greatest increase in the ROM (about 9.5 times after BPF and 1.83 times after UPF) (Figure 8f).

The model of the T2-L4 segments, fixed with the instrumentation, are shown separately (Figure 9).

Adjacent disc stress: The stress distributions are mostly concentrated at the lateral edge of IVD and were obvious in LB and RB (about 3 MPa in PREO, 6 MPa in UPF, and 15 MPa in BPF). The maximum stress was exhibited in the red regions, which was less than the average stress observed in the green regions (about 1.25 MPa~2.50 MPa in PREO, 2.50 MPa~3.50 MPa in UPF, and 6.25 MPa~8.75 MPa in BPF).

ROM: The maximum IVD ROM (T1/T2) in the three groups was 0.01 rad, 0.06 rad, and 0.18 rad, respectively, and displayed the following trend: BPF > UPF > PREO. Following GRT, the SAS ROM showed an obvious increase, especially in the LB and RB loading mode.

Reverse torque could be defined as the reaction of the C6 vertebra after the application of angular displacement, in order to measure the stability of the spine. At the same angular displacement loading level, a higher counter-torque value represented a smaller activity reaction in the spine. After UPF and BPF, the counter-torque decreased slightly, when the superior fixed segments moved down (T1-L4 fixed > T2-L4 fixed > T3-L4 fixed > PREO) (Figure 10).

### 3.3. Overall ROM of Spine and Bracing Force of Growing Rod in Three Growth Phases

Understanding the effect of UPF and BPF on the movement of the spine at different growth phases is vital. A significant increasing tendency was noticed among different loading modes for the ROM of three operations. The maximum ROM was exhibited at three surgical extensions (about 55°, 69° and 90° for UPF and about 56°, 70° and 87° for BPF). The average ROM of UPF was relatively large, compared to than that of BPF (Figure 11).

Bracing force is defined as an index that measures whether the rod will break easily. In this paper, we recorded the bracing force of the growing rod at three growth phases (Figure 12). The growing rod was braced six times, each at a distance of 2 mm. We can conclude that spines with smaller Cobb angles require more force to brace the same distance, with the third extension exhibiting a higher slope. Additionally, the UPF force was obviously greater than that of BPF. After the third surgery, the UPF bracing force reached 490 N.

## 4. Discussion

The growing rod technique (GRT) is usually an invasive surgical method that is used to treat EOS, which enables the correction of the coronal plane deformity, as well as the restoration of a well-balanced sagittal profile. The GRT procedure is often accompanied by a high risk of rod breakage and proximal junctional kyphosis (PJK), due to disrupted original structural balance after instrument implantation, creating unique biomechanical challenges [28,29,30].

Previous biomechanical studies demonstrated improved primary stability and reduced instrumentation loads when using multi-rod constructs [31,32,33]. However, they focused on loads supported by posterior instrumentation in relation to rod breakage, neglecting the changes incurred to the spinal biomechanical environment, incurred by the instrumentation, which is fundamental for the evaluation of the postoperative effect and to promote the patient’s life. While FEM may offer a convenient and simple way to describe the actual loads on spinal segments and instrumentation, an incomplete spinal environment construction may be a restriction for current studies [9,10]. The present study aimed to investigate the kinematic and biomechanical properties occurring after GRT across the long spinal structure and the posterior instrumentation and to determine the effects caused by unilateral or bilateral fixation in three growth phases. To achieve this, long-term spinal segment structures in the three growth phases were reconstructed to measure the stress and ROM distribution, counter-torque of C6, and the bracing force of the growing rod.

There are two things to note about the material properties and loading conditions. First, in this study, bony structures were assumed as linear elastic materials, without considering plastic deformation. Although there is overwhelming evidence that younger immature bone can undergo more plastic deformation prior to fracture than mature bone, because younger immature bone has a larger proportion of immature to mature enzymatic crosslinks [34,35], the yield stress of pediatric bone still remains at a high order of magnitude (about 100 MPa reported in [36]), which is not likely to be achieved in clinical surgery. Thus, we believe that the material property of linear elasticity is reasonable to a certain extent. Second, in the section of the following load, only the vertebral weight of T1-S1, as a percentage of body weight (BW), was defined in the literature [24], and the vertebral weight of T1 as a percentage of BW was set to (1.1 + 8% of head weight). According to the distribution rules, the average percentage of the vertebral weight of C1-C7 to BW was defined as approximately 1.1%. Thus, the percentages of the vertebral weight of C6 and C7 to BW were 6.9% and 1.1%. Additionally, with regard to T3-T5, the weight of superior limbs was taken into account on the basis of EI-Rich et al.’s study [37], resulting in the percentage of BW and following load of T3-T5 being obviously larger than those of T2 and T6.

The results demonstrated that BPF is more effective in reducing ROM than UPF (91.89% declines relative to PREO). In the meantime, SAS compensation is higher after BPF (increased significantly, even 32.5 times more than PREO). Compared to other studies on the surgical effect of BPF and UPF, UPF is mostly prone to causing thoracic abnormality, related to its instability [38,39]. However, stability cannot be neglected, as the protection of fixed segments is also an indicator to evaluate the effectiveness of the surgical results. The BPF surgical method should be considered first for patients with severe EOS. During this period, more attention must be paid to how to slow or stop the development of scoliosis. In addition, the prevention of postoperative infection and regression is also required. This is because the stress concentration phenomenon of SAS is obvious after BPF. The maximum counter-torque of the C6 segment after BPF also confirmed this point.

Additionally, PJK was able to describe sagittal thoracic deformity after posterior spinal fixation, which frequently occurs in SAS; this can cause local stress concentration and postoperative infection. Researchers compared the correction rate and complications of BPF and UPF by collecting the clinical data of 28 patients with EOS and found the BPF was significant in maintaining spinal stability but resulted in more postoperative infections [40,41]. Relatively, patients with a smaller Cobb angle are able to maintain higher levels of athletic ability after UPF in the case of relieving disease. These procedures result in a small risk of postoperative infection and a prolonged recovery time. This should encourage us to think carefully about the way we operate. The impact of different surgical procedures on children’s quality of life is different.

The effect of growth phases on the spinal movement characteristics and the bracing force of the growing rod was investigated. Accordingly, this may demonstrate that the risk of rod fractures becomes greater as the spine itself grows. Specifically, the unilateral growing rod is under more pressure compared to the bilateral growing rod. It is worth noting that, although the bracing force is linearly related to the bracing distance, the change in the Cobb angle will become insignificant in the subsequent growth phase. The bracing number should perhaps be reduced when the Cobb angle reaches a certain angle, allowing children to avoid undergoing multiple surgeries and damage associated with those surgeries. Furthermore, the bracing force of BPF is larger than that of UPF, achieving the largest 490 N in the third extension. Previous studies comparing the bracing force of the growing rod in 10 patients with EOS, when the growing rod was extended, have suggested that the best average peak force is 485 N [42]. The growing rod is prone to breaking after undergoing several surgical extensions; thus, maintaining an optimal extension distance may be key in reducing the risk of rod break. First, the rod integrated the fixed segments into a whole, with the fixed segment being able to bear more pressure due to the superior material properties, thus, adding to the stability of the whole spinal structure being maintained and the increased compensation and pressure in SAS being inevitable. Additionally, the larger numbers of fixed segments created more obvious restrictions on spinal activity. The incidence of PJK would be greatly increased if the SAS was located at the apex of the spinal sagittal curvature. Meanwhile, the bracing force of the growing rod was mostly influenced by the muscle force, of which the vertical component increased as the Cobb angle decreased.

Due to the limitation of no corresponding in vitro experimental results to assist with verification, the spines from three growth phases were reconstructed with 11 models that were used to perform finite element analysis. It is difficult to obtain a long segmental specimen from a patient with EOS. Secondly, there are almost no animal specimens, let alone specimens with a curve that is the same as the one that we constructed, and it is not suitable to use humans to verify the results. Finally, it is hard to provide specimens from different growth phases, regardless of whether human specimens or animal specimens are used. As things stand now, FEM may be the best way to study the biomechanical mechanism of EOS.

## 5. Conclusions

This paper investigated the kinematic and biomechanical properties that occur after GRT, across the long spinal structure and the posterior instrumentation, and how they are affected by unilateral or bilateral fixation. Compared to previous studies, the models that we constructed are more representative of the true spinal environment, with six kinds of surgical procedures being simulated throughout three growth phases, using eleven established models. Our results show that BPF is more stable than UPF at the expense of more SAS compensations, especially when the superior fixed segment is closer to the head, placing more pressure on SAS. Meanwhile, the growth of the spine itself may be another interpretation for the higher risk of rod fractures. In summary, the spinal curvature of the patient should be considered when selecting the surgical procedure, as should the patient’s therapy periods, as well as the surgically fixed segment. The next avenue of future work could be adding the corresponding in vitro experiment to verify our results.

## Figures and Tables

**Figure 1 ijerph-19-02057-f001:**
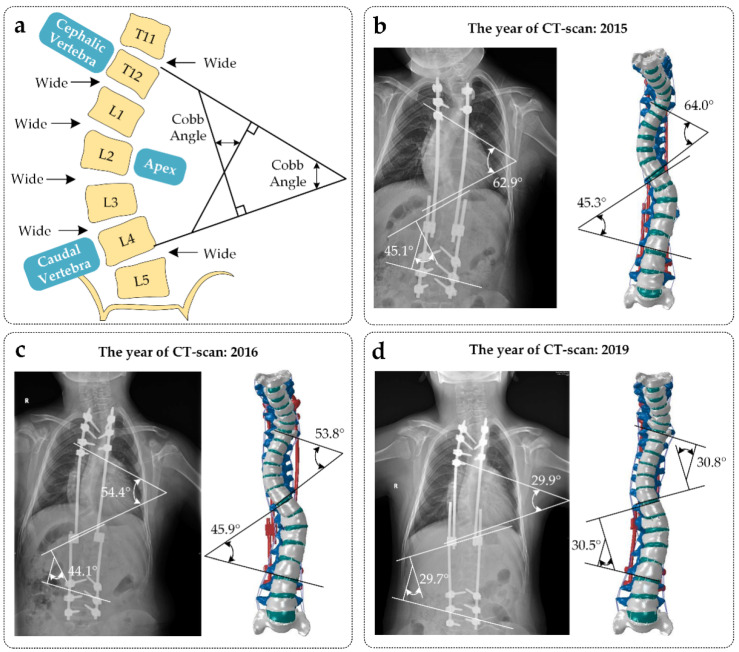
(**a**) Definition method of Cobb angle. The Cobb angle is the angle of intersection between the vertical of the superior edge of the cephalic vertebra and the vertical of the inferior edge of the caudal vertebra. (**b**–**d**) contain the CT taken after the patient’s first surgery (2015), the patient’s second surgery (2016), and the patient’s third surgery (2019). The reconstructed model conforms to the CT images.

**Figure 2 ijerph-19-02057-f002:**
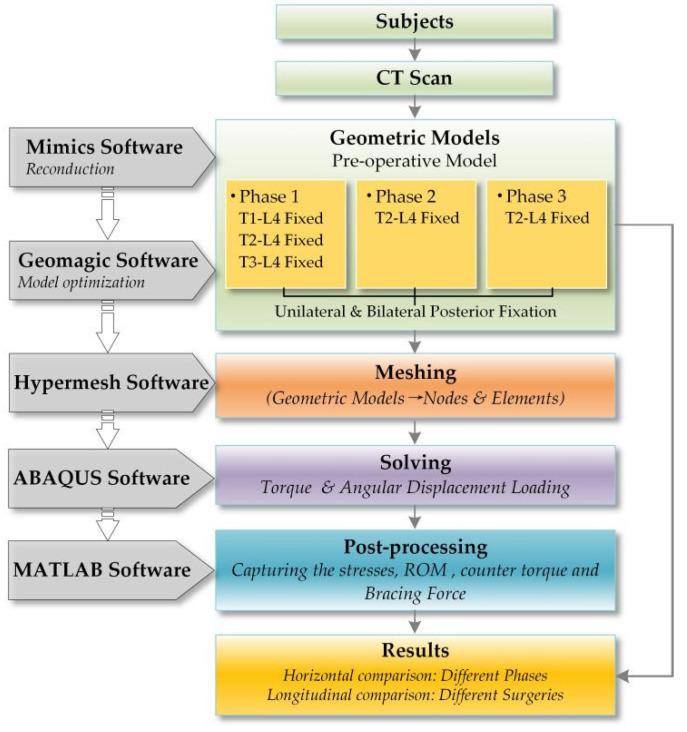
Flowchart of the entire study.

**Figure 3 ijerph-19-02057-f003:**
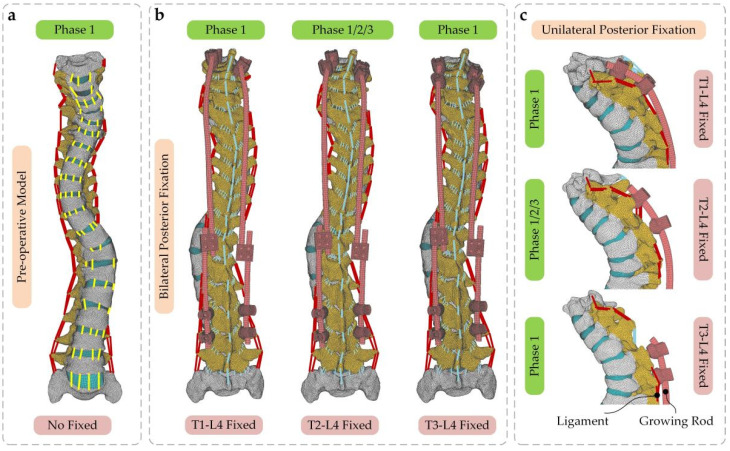
(**a**) Front view of the pre-operative model in growth phase 1; (**b**) rear view of the post-operative models with three fixed modes following BPF. The model fixed on T2-L4 segments was reconstructed to represent the three growth phases; (**c**) side view of the post-operative models with three fixed modes following UPF. The model fixed on T2-L4 segments was reconstructed to represent the three growth phases.

**Figure 4 ijerph-19-02057-f004:**
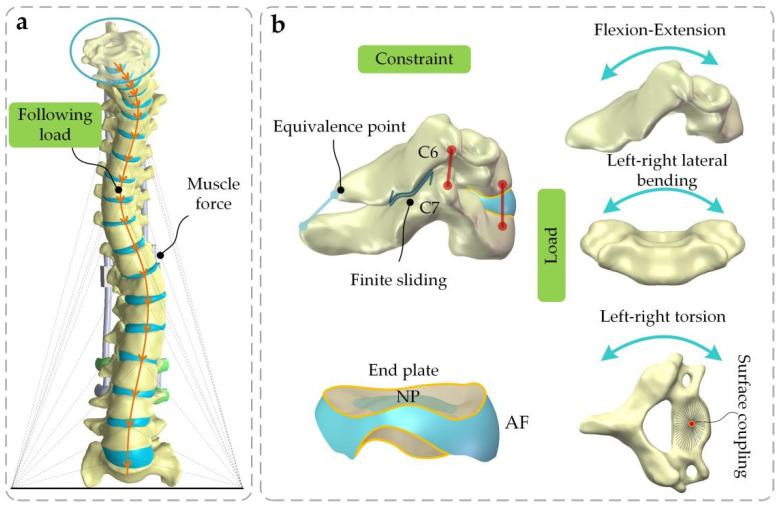
(**a**) Muscle force and pressure on the vertebra can be converted into the following loads when passing down from a direction perpendicular to the surface of each vertebra [23]. (**b**) Face-to-face binding was used between vertebra and IVD, and finite sliding was utilized between facet joints. Six directional torque loads were applied in the superior extremity of C6: Flex, Ext, LB, RB, LT, and RT, following the vertebral surface coupled to a point.

**Figure 5 ijerph-19-02057-f005:**
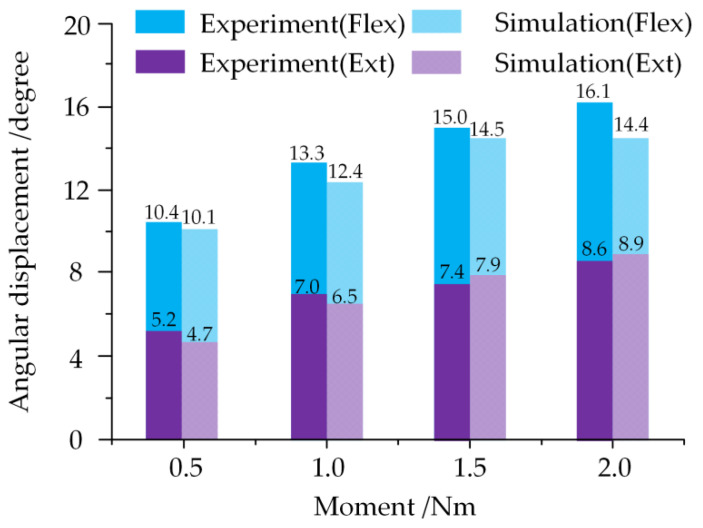
Comparison of angular displacement between experimental results and simulated results [27].

**Figure 6 ijerph-19-02057-f006:**
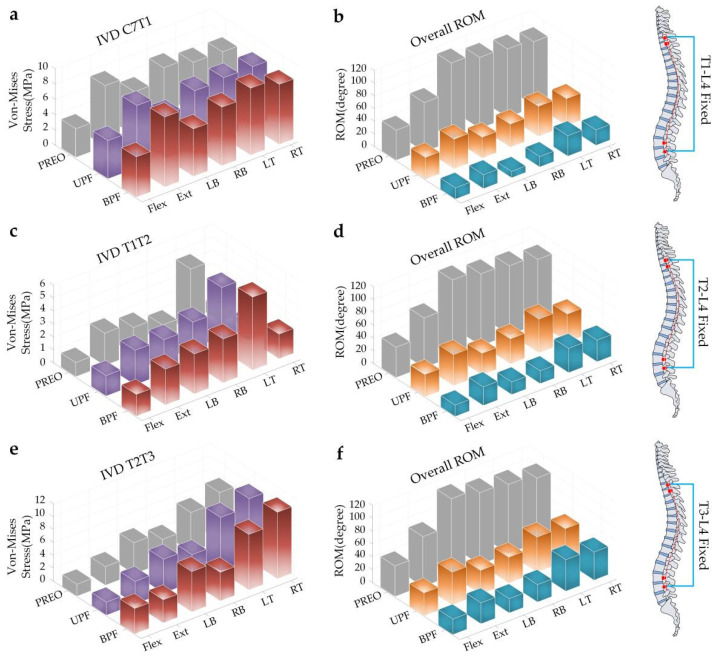
SAS IVD stress and overall ROM of different surgical procedures under torque loading. (**a**) The stress changes of IVD C7T1; (**b**) The overall ROM changes after T1-L4 fixed; (**c**) The stress changes of IVD T1T2; (**d**) The overall ROM changes after T2-L4 fixed; (**e**) The stress changes of IVD T2T3; (**f**) The overall ROM changes after T3-L4 fixed.

**Figure 7 ijerph-19-02057-f007:**
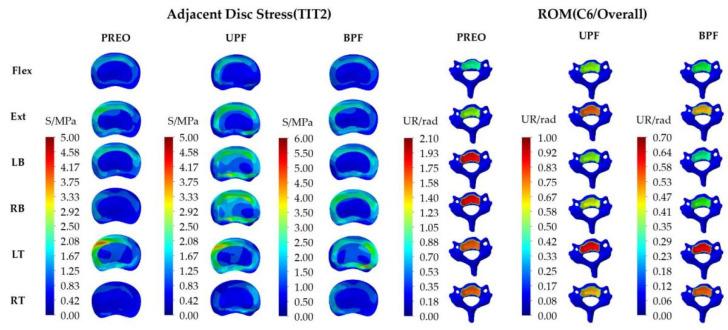
IVD stress distribution of SAS and ROM distributions in the C6 segment. ROM has been defined as the different values between the inferior segment and superior segment, thus, the overall ROM is the same as the ROM of C6.

**Figure 8 ijerph-19-02057-f008:**
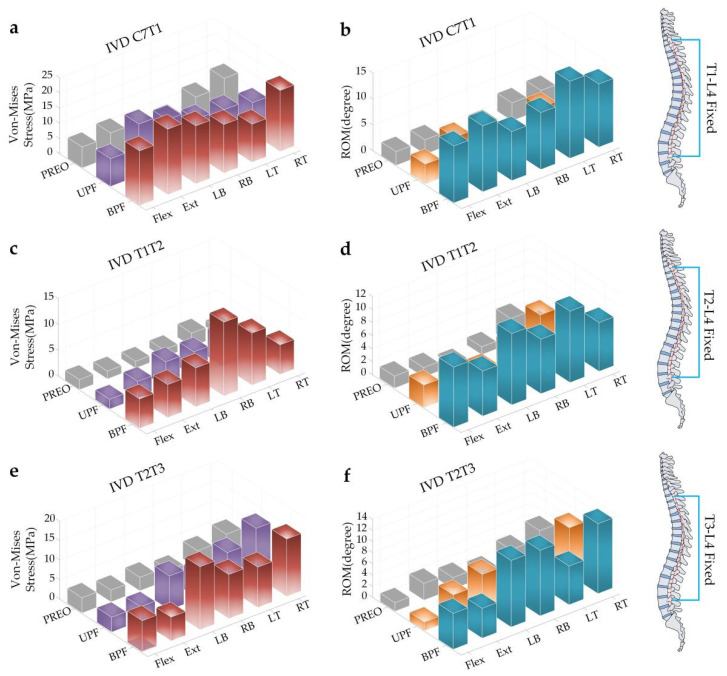
Stress and ROM of IVD in SAS with different surgical procedures under angular displacement. (**a**) The stress changes of IVD C7T1; (**b**) The ROM changes of IVD C7T1; (**c**) The stress changes of IVD T1T2; (**d**) The ROM changes of IVD T1T2; (**e**) The stress changes of IVD T2T3; (**f**) The ROM changes of IVD T2T3.

**Figure 9 ijerph-19-02057-f009:**
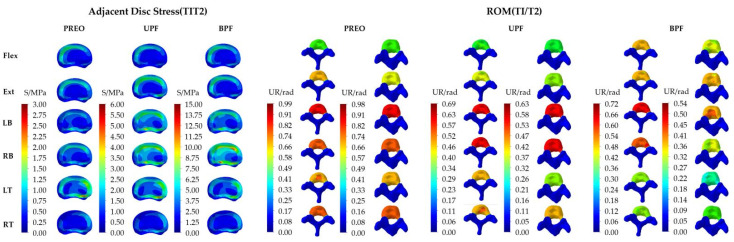
IVD stress distribution and T1/T2 ROM distribution in SAS. The ROM of IVD (T1/T2) was defined as the different values between T1 and T2.

**Figure 10 ijerph-19-02057-f010:**
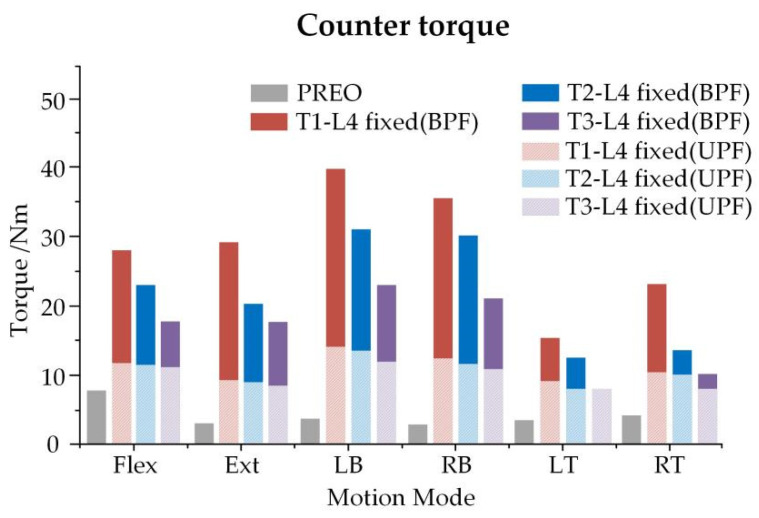
Counter torque on C6 in the different surgical procedures.

**Figure 11 ijerph-19-02057-f011:**
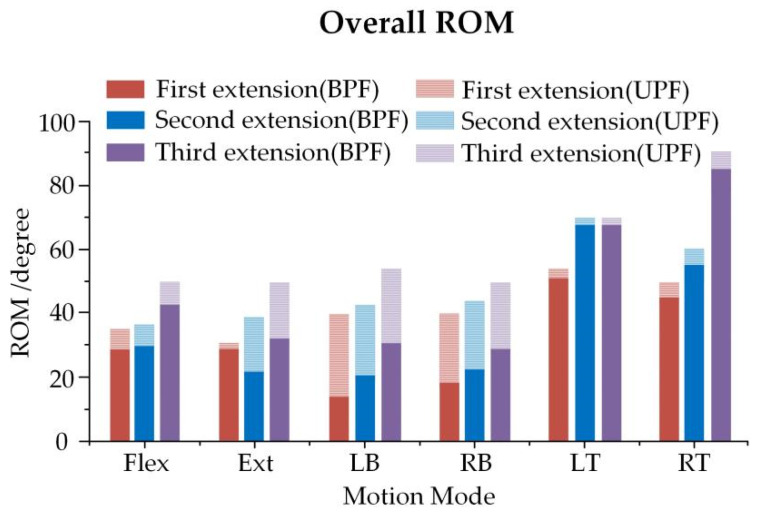
Overall ROM of the spine in the three growth periods.

**Figure 12 ijerph-19-02057-f012:**
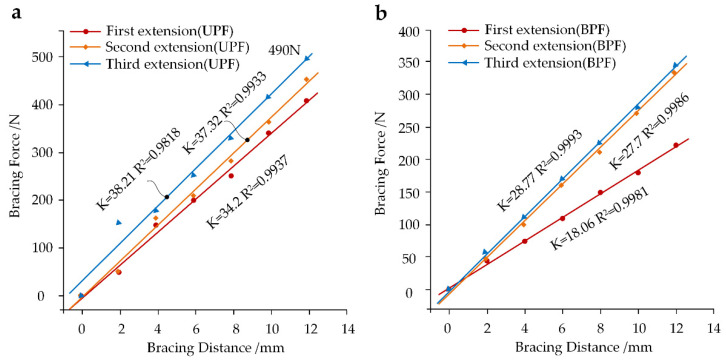
The bracing force of the growing rod in three surgical procedures. (**a**) The liner relation between bracing distance and bracing force after UPF; (**b**) The liner relation between bracing distance and bracing force after BPF. These data were normally distributed and significantly correlated at 0.01 level according to the Shapiro–Wilk test (*p* > 0.05) and Pearson correlation. K represents the slopes of the curves and R^2^ represents the regression coefficients of bracing force against the bracing distance.

**Table 1 ijerph-19-02057-t001:** Material properties in the present finite element models.

Element Construction	Element Type	Elasticity Modulus/MPa	Poisson Ratio	Cross-Sectional Area/mm^2^	Scale Factors	Reference
Cortical bone	Hexahedron	1.344 × 10^4^	0.30	—	—	[16,17]
Cancellous bone	Quadrilateral	2.41 × 10^2^	0.30	—	0.805 (b)	[18]
Posterior	Hexahedron	3.5 × 10^3^	0.25	—	(a)	[19]
Endplate cartilage	Quadrilateral	2.38 × 10	0.40	—	(a)	[20]
Nucleus pulposus	Hexahedron	1.0	0.49	—	(a)	
Annulus fibrosus	Hexahedron	4.2	0.45	—	0.782 (b)	
ALL	Three-dimensional truss	7.8	0.12	63.7	0.893 (b)	[20]
PLL		1.0 × 10	0.11	20.0		
ITL		1.0 × 10	0.18	1.80		
CL		7.5	0.25	30.0		
ISL		8.0	0.14	30.0		
SSL		1.0 × 10	0.20	40.0		
LF		1.5 × 10	0.062	40.0		
Growing rod	Hexahedron	1.1 × 10^5^	0.30	—	(a)	[21]

(a) Indicates that the material parameters were the same as the values of an adult. (b) The scale factors were used to scale adult material parameters to child ones [22].

**Table 2 ijerph-19-02057-t002:** Following load magnitude of each segment of C6-S1 [24].

Vertebral	Percentage of BW (%)	Following Load(N)
C6	6.9	20.7
C7	1.1	3.3
T1	1.1	3.3
T2	1.1	3.3
T3	1.3 + 4.0 (Superior limbs)	15.9
T4	1.3 + 4.0 (Superior limbs)	15.9
T5	1.3 + 4.0 (Superior limbs)	15.9
T6	1.3	3.9
T7	1.4	4.2
T8	1.5	4.5
T9	1.6	4.8
T10	2.0	6.0
T11	2.1	6.3
T12	2.5	7.5
L1	2.4	7.2
L2	2.4	7.2
L3	2.3	6.9
L4	2.6	7.8
L5	2.6	7.8
S1	2.6	7.8

## Data Availability

The data are available upon request.

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
