# Peer review of "Effects of Growing Rod Technique with Different Surgical Modes and Growth Phases on the Treatment Outcome of Early Onset Scoliosis: A 3-D Finite Element Analysis"

_ijerph, 2022, doi:10.3390/ijerph19042057_

Round 1

Reviewer 1 Report

  1. There are several mistakes, the text is partially not understandable; the language must be better.  Extensive editing of English language and style required.
  2. I wonder if any addtional x-rays (of the patient) or ct-scans were done. If yes, the approval from local ethic committee is mandatory.
  3. in discussion: "growing rod technique (GRT) is usually an invasive surgical method utilized to treat EOS, allowing the restoration of a well-balanced sagittal profile". I think is should be: "....allowing the coronal plane deformity correction as well as  the restoration of a well-balanced sagittal profile"
  4. in discussion: "The incidence of PJK would be greatly increased if the SAS is located at the apex of spinal curvature". It should look like: "The incidence of PJK would be greatly increased if the SAS is located at
    the apex of sagittal spinal curvature".

Reviewer 2 Report

Thank you for the privilege of reviewing the article on effects of growing rod technique with different surgical modes and growth phases on the treatment of early-onset scoliosis: a 3-D finite element analysis.

The article is interesting and describes a very interesting 3D finite element analysis of unilateral or bilateral posterior instrumentation of the spine in a child suffering from early onset scoliosis. I think this article should be considered for publication by a pediatric orthopedic journal.

The finite element analysis represents a highly sophisticated approach, but the discussion of the results obtained hereby is of low quality. I recommend you rewrite the discussion section and describe your findings and explain to readers of IJERPH.

The English of the article needs improvement. I recommend you revise the English of this article with support of a professional medical writing company.

Line 18: replace “Literatures” by “Several authors ”

Line 40: rephrase “During a period of growing rapidly …”.

Line 47: replace “choice of fixed mode” by “choice of the fixed mode of stabilization”.

Line 50: rephrase “is prone to happen phenomenon of fracture”.

Line 56: replace “Most past literature” by “Older publications”.

Line 57-58: rephrase “of how does spine response to instrumentation after UPF or BTF”.

Line 76: rephrase “the occurrence mechanism”.

Line 85: rephrase “is with no other spinal disease”.

Line 90: replace “was” by “is”.

Line 157: rephrase: “each vertebral following load”

Line 163: rephrase “Define the expansion coefficient”.

Line 173: Give reference for “Panjabi”.

Line 179 Table 2: Please explain in the discussion section why the “Percentage of BW” and “Following load” differs so much between C6 and C/, T2 and T3, and T5 and T6.

Line 312-313: rephrase “By observing the ROM … related to …”.

Line 325-326: rephrase “causing … concentration”.

Line 335-336: rephrase “thus proving that … itself”.

Line 346: delete “We concluded … above”.

Line 358: rephrase “The difficulty lied”.

Line 359: rephrase

Line 375: delete: “This study can inspire surgeons to make surgical decisions.”

Reviewer 3 Report

The paper deals with the kinematic and mechanical properties occurring after GRT across the long spinal structure and the posterior instrumentation, as affected by unilateral or bilateral fixation. The paper does not deal with mechanical properties but only with kinematic properties. What is the take-home message that the authors want to provide? 
How their finding will be helpful? What will be the recommendation for the readers?
Several sentences are not written in proper English syntax making the reading difficult sometimes. 

Please see below the comments regarding the paper. 

Majors comments:

The authors could have used more recent biomechanical data for cortical bone in children. Furthermore, the authors do not take into consideration plastic deformation.
Please update the references and discuss that your model is not elastoplastic.

Emily Szabo, Clare Rimnac, Biomechanics of immature human cortical bone: A systematic review,
Journal of the Mechanical Behavior of Biomedical Materials, Volume 125,2022, 104889, ISSN 1751-6161

In vitro ultrasonic and mechanic characterization of the modulus of elasticity of children cortical bone
JP Berteau, C Baron, M Pithioux, F Launay, P Chabrand, P Lasaygues
Ultrasonics 54 (5), 1270-1276

Ratio between mature and immature enzymatic cross-links correlates with post-yield cortical bone behavior: An insight into greenstick fractures of the child fibula
JP Berteau, E Gineyts, M Pithioux, C Baron, G Boivin, P Lasaygues, ...
Bone 79, 190-195

The authors used the data of Panjabi, without clearly making the references to the paper. 
White, A A I, Panjabi M M. Clinical biomechanics of the spine. Philadelphia. 1992.

There is no most recent reference related to children's spines? 
Please provide more references for the validation. 

For instance, look at this one:
Lopez-Valdes FJ, Lau S, Riley P, Lamp J, Kent R. The biomechanics of the pediatric and adult human thoracic spine. Ann Adv Automot Med. 2011;55:193-206.

Please test if the data are normally distributed using the Shapiro Wilk test. In case of a normal distribution, please perform a Pearson correlation ( the one probably depicted there); otherwise, please perform a Spearman correlation. What means the K on the graph? If this is the K index theory, please elaborate. 

Minors comments:

Thoracic curve Cobb angle of 62.9°/54.4°/29.9°, please precise what the number means, the APEX level. Please provide 2D pictures that depict the Cobb Angle.
Please provide a clear timeline of the 11 CT-scan performed (a timeline will help understand). 
Please elaborate on the functional spine units used for validation, and this part is not clear.

Round 2

Reviewer 2 Report

I think the manuscript has been improved significantly, congratulations to the authors. I have only 2 minor recommendations:

Line 121: replace "evaluated" by "evaluate"

Line 377: replace: "Percentage of BW and Following" by "percentage of BW and following"

Reviewer 3 Report

The authors answered most of my comments. One last mistake on table 1

please delete the b for cortical bone since you used actual values from children's bone. 

Furthemore, there is no caption for it. 
